# Antimicrobial and Antibiofilm Effect of 4,4′-Dihydroxy-azobenzene against Clinically Resistant Staphylococci

**DOI:** 10.3390/antibiotics11121800

**Published:** 2022-12-11

**Authors:** María Pérez-Aranda, Eloísa Pajuelo, Salvadora Navarro-Torre, Patricia Pérez-Palacios, Belén Begines, Ignacio D. Rodríguez-Llorente, Yadir Torres, Ana Alcudia

**Affiliations:** 1Departamento de Química Orgánica y Farmacéutica, Facultad de Farmacia, Universidad de Sevilla, c/Profesor García González, 2, 41012 Sevilla, Spain; 2Departamento de Microbiología y Parasitología, Facultad de Farmacia, Universidad de Sevilla, c/Profesor García González, 2, 41012 Sevilla, Spain; 3UGC Enfermedades Infecciosas, Microbiología Clínica y Medicina Preventiva, Instituto de Biomedicina de Sevilla IBIS, Hospital Universitario Virgen Macarena, CSIC, Universidad de Sevilla, 41009 Seville, Spain; 4Departamento de Ingeniería y Ciencia de los Materiales y del Transporte, Escuela Politécnica Superior, Universidad de Sevilla, Virgen de África 7, 41011 Sevilla, Spain

**Keywords:** *Staphylococcus aureus*, *Staphylococcus pseudintermedius*, azo compounds, oxidative stress, biofilms, clinical samples, cutaneous plasters

## Abstract

The spread of antibiotic resistance among human and animal pathogens is one of the more significant public health concerns. Moreover, the restrictions on the use of particular antibiotics can limit the options for the treatment of infections in veterinary clinical practice. In this context, searching for alternative antimicrobial substances is crucial nowadays. In this study, 4,4′-dihydroxy-azobenzene (DHAB) was tested for its potential in vitro as an antimicrobial agent against two relevant human and animal pathogens, namely *Staphylococcus aureus* and *Staphylococcus pseudintermedius*. The values of minimal inhibitory concentration (MIC) were 64 and 32 mg/L respectively, and they comparable to other azo compounds of probed antimicrobial activity. In addition, the minimal bactericidal concentrations (MCB) were 256 and 64 mg/L. The mechanism by which DHAB produces toxicity in staphylococci has been investigated. DHAB caused membrane damage as revealed by the increase in thiobarbituric acid reactive substances (TBARS) such as malondialdehyde. Furthermore, differential induction of the enzymes peroxidases and superoxide dismutase in *S. aureus* and *S. pseudintermedius* suggested their prevalent role in ROS-scavenging due to the oxidative burst induced by this compound in either species. In addition, this substance was able to inhibit the formation of biofilms by both bacteria as observed by colorimetric tests and scanning electron microscopy. In order to assess the relevance of DHAB against clinical strains of MRSA, 10 clinical isolates resistant to either methicillin or daptomycin were assayed; 80% of them gave values of CMI and CMB similar to those of the control *S. aureus* strain. Finally, cutaneous plasters containing a composite formed by an agar base supplemented with DHAB were designed. These plasters were able to inhibit in vitro the growth of *S. aureus* and *S. pseudintermedius*, particularly the later, and this suggests that this substance could be a promising candidate as an alternative to antibiotics in the treatment of animal skin infections, as it has been proven that the toxicity of this substance is very low particularly at a dermal level.

## 1. Introduction

In the last few decades there has been a notorious emerging problem of infections caused by multiresistant microorganisms [1]. The increasing prevalence of infections produced by multidrug-resistant bacteria is rendering a growing number of antibiotics ineffective and has highlighted the necessity for developing alternatives [2,3] including (genetically-modified) bacteriophages, lysins, antimicrobial peptides, antibodies, and new antimicrobial synthetic nanochemicals [4,5,6]. For instance, steroids derivatives have recently been suggested as alternative antimicrobial substances against methicillin-resistant *S. aureus* [7]. The mechanisms of action of some of these new substances including metallic nanoparticles, carbon nanostructures, antimicrobial peptides, etc., are being deciphered [5,6,8].

In addition to this problem in human health, the restrictions on veterinarians accessing to latest generation of antibiotics—whose use is normally hospital-restricted to limit the spread of resistance—presents a very complicated scenario at the time of treating these infections [9,10]. In fact, *S. pseudintermedius*, followed by *S. aureus* strains, is responsible for most of the skin infection cases observed in animals and is being envisioned as a significant and critical health problem due to its zoonotic potential [11,12,13,14]. In this sense, wild animals are considered as reservoirs and sentinels of methicillin-resistant *S. aureus*, a problem that must be envisioned from the One Health perspective [15]. Consequently, the use of topical non-antibiotic treatments for cutaneous application in veterinary dermatology could be an outstanding alternative to avoid this problem [4].

Azo compounds are chemical entities bearing an azo moiety R-N=N-R′ as a part of the molecular structure in which R and R′ can be either aryl or alkyl, however the most stable derivatives contain two aryl groups [16]. Synthetic azo dyes are widely used in different industries and, due to extensive electron delocalization, have vivid colors such as reds, yellows, purples, or oranges, and they are usually employed as synthetic colorants for textiles, paints, food, or cosmetics in different industries [16,17]. Moreover, azo compounds have been extensively used in other areas as indicators in sensor applications with fluorescence or optoelectronic-related functional materials [18,19].

Moreover, they have multiple biomedical applications [20,21,22]. In this sense, in the second quarter of the 20th century, the antimicrobial properties of azo dyes were discovered [23]. Furthermore, the pharmacological activity of azo derivatives has been widely demonstrated: (a) against *Escherichia coli*, *Listeria monocytogenes,* and *Staphylococcus aureus* they have shown excellent antibacterial activities using halogenated azo-aspirin. In addition, computational studies concerning Prontosil (sulfamidochrysoidine, one of the first antimicrobial drugs) have been developed to demonstrate a relatively broad effect against Gram-positive cocci but unfortunately not against Enterobacteriaceae [24,25,26]; (b) compounds such as Evans blue or Congo red are both inhibitors of the HIV virus, since they are able to bind to protease and reverse transcriptase [27]; (c) 1,3,4-oxadiazoles have activities as potent antifungal, analgesic, antimicrobial, antituberculosis, or cytotoxic agents [22]. Among the most recognized are the commercially available heterocyclic triazole-based derivatives such as ketoconazole or fluconazole drugs, commonly used in medicine as potent antifungals [28]. In addition, azo compounds can bind to numerous metals forming complexes that exhibit antifungal and antibacterial activities through DNA inhibition-related mechanisms [29].

With regard to their action, Gerhardt Domagk established that the antimicrobial properties of Prontosil were due to the reduction of the azo bond thus rendering the toxic product sulfanilamide [30]. Indeed, diverse microorganisms can perform the reduction of the azo bond, including human and animal intestinal microbiota, skin microbiota, and environmental microorganisms [31]. Reduction of the azo bond is performed by azoreductases, a group of diverse FAD-containing enzymes that use NADH or NADPH as electron donors and can perform reduction in aerobic or anaerobic conditions, rendering the corresponding aromatic amines [32]. 

In spite of the abundant literature concerning the antimicrobial properties of azo compounds [33,34,35,36], much less is known about the effect of these compounds on biofilm formation by pathogenic bacteria [37,38] as well as the exact mechanisms by which these substances exert their toxicity in bacteria. In particular, azo dyes are known to induce oxidative stress in animals [39,40] and plants [41] as well as in Gram-negative bacteria [42] as deduced from increased levels of ROS-scavenging enzymes. Moreover, they induce damage in the membrane and some of them can also display genotoxicity [43,44].

In this context, the aims of this study were: (1) establishing the antimicrobial properties of 4,4′-dihydroxyazobenzene (DHAB) towards two staphylococci of human and veterinary importance, namely *S. aureus* and *S. pseudointermedius*; (2) elucidating the toxicity mechanisms; (3) investigating the putative effect of 4,4′-dihydroxyazobenzene on the formation of biofilms; (4) testing the antimicrobial activity of 4,4′-DHAB against ten *S. aureus* isolates of clinical relevance; and (5) designing plasters containing 4,4′-DHAB for presumptive therapeutic applications in dermal veterinary infections.

## 2. Results and Discussions

### 2.1. Determination of Bacterial Susceptibility towards DHAB

The prevalence of antibiotic multiresistant bacteria is one of the more significant problems and highlights the urgent need for alternatives to antibiotics [2,3]. Methicillin-resistant *S. pseudintermedius* and *S. aureus* (MRSA) cause most of the cutaneous infections in pets and can be transmitted to humans, with the number of zoonotic infections caused by these pathogens raising [14,15]. In particular, methicillin-resistant *S. aureus* is one of the most prevalent human and animal pathogens [15,45]. Moreover, *S. pseudintermedius* causes the majority of skin infections in pets and can be also transmitted to humans as a zoonotic pathogen, with many of the isolates being methicillin resistant [11,12,13,14,46,47]. Moreover, the appearance of multiresistant strains and limitations on the use some antibiotics reserved for only human use due to precautionary principles sometimes makes the treatment of infections in animals challenging [48,49].

In this sense, azo compounds are known to inhibit bacterial growth, particularly in Gram-positive bacteria [25,26]. In this study, the azo compound DHAB has been tested as an antimicrobial compound. A preliminary test using paper disks loaded with DHAB was carried out. The results showed that both staphylococcal strains showed certain susceptibility towards DHAB (Figure 1), depicting the higher susceptibility of *S. pseudintermedius* compared to *S. aureus*. The diameters of the halos were 17 mm for *S. aureus* and 21 mm for *S. pseudintermedius*. The same test was performed for two Gram-negative bacteria, i.e., *Escherichia coli* ATCC 25922 and *Pseudomonas aeruginosa* ATCC 27853. No inhibition halos were observed (not shown) so the study with Gram-negative strains was not continued further.

In a quantitative approach, both the MIC and the MBC were determined. There was a good correlation between the results previously obtained by visual inspection of the inhibition halos and the data of MIC. *S. pseudintermedius* was the most sensitive strain, showing an MIC value of 32 mg L^−1^, whereas *S. aureus* showed an MIC value of 64 mg L^−1^. The MBC was reached at 64 and 256 mg L^−1^ for *S. pseudintermedius* and *S. aureus*, respectively (Table 1).

These results indicate that DHAB showed good antimicrobial properties against *S. pseudintermedius* and *S. aureus*, particularly against the former one. Furthermore, DHAB was also tested against *Escherichia coli* and *Pseudomonas aeruginosa* but this compound seems not to be useful for Gram-negative strains (results not shown). In our case, growth of *E. coli* and *P. aeruginosa* in the presence of DHAB as the sole carbon source was observed (Appendix A), indicating that these Gram-negative bacteria can degrade DHAB and use it as metabolic source of carbon and energy. In fact, some bacteria can not only tolerate but also degrade and even fully mineralize azo dyes in waste waters in aerobic, anaerobic, or sequential conditions, with them being useful for the bioremediation of these substances [50,51].

Our results are in accordance with the previous data of other azo compounds with comparable structures (Figure 2), thus indicating that these substances can effectively inhibit the growth of Gram-positive bacteria, but they are not useful towards Gram-negative bacteria (Table 2).

As a matter of comparison, the values of MIC for several antibiotics for the type strain *S. aureus* ATTC29213 can be visualized in Table 3. These data are available from EUCAST (European Committee on Antimicrobial Susceptibility Testing) [52]. The value of MIC for DHAB is higher (4- to 20-fold) than that reported for antibiotics. However, the same situation occurs with other azo compounds of similar structure which have recognized antimicrobial activity [25,35].

### 2.2. Evaluation of Oxidative Stress

In order to obtain insights into the antimicrobial mechanism of DHAB, the activity of some antioxidant enzymes involved in scavenging reactive oxygen species (ROS) was measured, namely, catalase, total peroxidases, and superoxide peroxidase (Table 4). Our results indicated that the activity of catalase suffered minor changes without a constant trend in relationship to the presence of DHAB and the dose. Regarding SOD and total peroxidases, a different behavior was observed depending on the species: in *S. aureus* total peroxidases seem to have a preponderant role in ROS detoxification, whereas in *S. pseudintermedius*, particularly at high concentrations (MBC), SOD seemed to be the enzyme most involved in ROS scavenging.

The determination of ROS-scavenging enzymes showed that there was no apparent induction of catalase. Similar results were found in plants of *Allium cepa*, whose catalase activity was even inhibited by azo dyes [41]. In contrast, in the case of *S. pseudintermedius*, which was the most susceptible strain, SOD activity was enhanced by 200%. In this regard, SOD seemed to play a prevalent role in detoxifying ROS produced upon exposure to DHAB in this species. In fact, SOD has been suggested as a marker of oxidative stress caused by the azo dye Acid Black in plants [41] and animals such as the fish *Labeo rohita* [53]. In addition, in *S. aureus*, total peroxidases seem to be the enzymatic activities most induced upon exposure to DHAB. In previous work exposing *S. aureus* to other azo dyes such as Orange II and Sudan III, it was concluded that differential gene expression was recorded in both cases, suggesting the different adaptation of the strain towards diverse azo dyes [54]. However, genes involved in stress response such as methionine-S-sulfoxide reductase—involved in tolerance to oxidative stress—were induced by both azo dyes [55]. In the Gram-positive *Lysinibacillus* sp. strain RGS, which is able to degrade Reactive Orange 16, the enzymes catalase and SOD seem to participate not only in the adaptation to oxidative stress but in the degradation of the dye, together with some other reductases [56]. Moreover, in a recently reported study, five different azo dyes derived from stilbene were synthetized. Several of these substances had antioxidant activity but only weak antimicrobial activity against *Pseudomonas aeruginosa* and *Streptococcus pneumoniae* [36].

Given that oxidative burst was detected, the possibility that this compound could affect the stability of the membranes due to lipid peroxidation has been explored. The results of the determination of thiobarbituric acid reactive substances (TBARs), mainly malondialdehyde (MDA), showed great increases in the presence of DHAB in a dose-dependent manner (Figure 3): MDA content was enhanced by 12 and 11 folds in *S. pseudintermedius* and *S. aureus*, respectively, at MIC. At MBC the increases were 20 and 37 folds, respectively. These results indicate that the presence of DHAB compromises the stability of the membrane.

One of the effects of oxidative stress at the cellular level is the compromise of membrane stability. The formation of ROS leads to lipid peroxidation and affects the structure, stability, and function of the cell membrane [57]. Our results show great increases in MDA levels, a marker of lipid peroxidation and damage to membranes, which revealed damage to the cell membranes of bacteria according to data reported for azo dyes for human intestinal microbiota [58]. Moreover, recent studies have shown deep changes in the transcriptome of *S. aureus* exposed to another azo dye, namely Sudan III. Particularly affected was the expression of genes involved in cell wall/membrane biogenesis and biosynthesis, suggesting that Sudan III damages the bacterial envelopes [54]. Further analysis indicated that most of them codified membrane transporters.

### 2.3. Effect of DHAB on Biofilm Formation

One important trait concerning the pathogenicity of bacteria is their ability to form biofilms [59,60]. Since the formation of biofilms is a pathogenicity factor of these bacteria, in particular of *S. pseudintermedius* [61,62], the possibility that these compounds could block or minimize the formation of biofilms has been explored. The effect of DHAB on biofilm formation in vitro was evaluated in microtiter plates at base two logarithm concentrations of these compounds (Figure 4). Concerning *S. pseudintermedius*, the results showed that, besides inhibiting bacterial growth, DHAB also prevented biofilm formation by this bacterium according to concentration. Thus, the formation of the biofilm was severely affected at 4 µg mL^−1^ and it was fully inhibited at 16 µg mL^−1^ (Figure 4A). Hence, in the case of *S. aureus*, the concentrations of DHAB needed to affect biofilm formation were higher than in the case of *S. pseudintermedius*: 16 µg mL^−1^ to affect biofilm formation and 64 µg mL^−1^ to fully inhibit this process (Figure 4). Taking into account that biofilm formation is a major virulence factor in both *Staphylococcus* species, the inhibitory effect of DHAB on biofilm formation is clearly desirable.

The formation of biofilms in the presence of the antimicrobial compounds has been analyzed by SEM (Figure 5). As shown in Figure 5A,B,E,F, both bacteria were able to multiply on and attach to the surface of the glass slides, where large colonies were observed. These colonies showed the presence of a great amount of extracellular material in the form of particulate material and fibers that probably facilitate the attachment of bacteria to the surface for the formation of the biofilm (Figure 5B,F). In the presence to DHAB at concentration MIC, the formation of the biofilm was severely affected and only small and disperse colonies were observed for both bacteria (Figure 5C,G). In a dose-dependent manner, when DHAB was used at MBC, the number of bacteria attached to the glass surface was very low, no colonies were observed but only individual bacteria indicating that the multiplication of the bacteria was fully abolished (Figure 5D,H). Moreover, the deposition of extracellular material was inhibited, and the morphology of the cells was also affected, particularly in the case of *S. pseudintermedius* (Figure 5D), for which smaller and deformed cells were observed. In this regard, the effect of the compound is clearly beneficial since inhibition of biofilm was achieved even before cell growth inhibition. In addition, the resistance towards antibiotics is usually increased in bacterial cells embedded in a biofilm as compared to planktonic cells [63,64].

### 2.4. Testing DHAB against Clinical Isolates of S. aureus

Since DHAB was tested first against the control strain *S. aureus* ATCC29213, our aim was to evaluate the utility of this compound against clinically relevant strains isolated from different sources (blood, swabs, injuries, nasal smear, the respiratory tract, etc.) as shown in Table 5. The genetic variability of the isolates was determined by pulse field gradient electrophoresis (Figure 6). The results allow for grouping of the isolates into two main branches, comprising three and seven strains, respectively. Most of the isolates were resistant towards methicillin except for ST5 and ST30DR, which were methicillin sensitive (MSSA) but resistant towards daptomycin (DAPTO R).

The results of susceptibility towards DHAB showed that eight out of the ten isolates had similar MIC and MCB values to those previously determined for the control strain ATCC 29213, thus indicating the validity of the results for these clinically relevant isolates. Moreover, two of these isolates (namely SA2 and SA4) were methicillin sensitive but showed resistance towards daptomycin (DAPTO R). In both cases, the susceptibility of these two strains towards DHAB was also similar to that of ATCC 29213. In contrast, two methicillin-resistant isolates, namely clones SA9 and SA10, isolated from samples of a swab and the respiratory tract, respectively, showed values of MIC and MBC 4-fold higher than those determined for the control strain, suggesting in this case that DHAB would not be adequate as an antimicrobial for these particular isolates.

The utility of DHAB against 80% of clinical isolates is of the upmost importance. Real clinical isolates can display different biochemical characters and antibiotic susceptibility as compared to the type strain *S. aureus* ATCC29213 [65]. Of significant relevance are clinical isolates resistant to methicillin and daptomycin [45] owing to the difficulty of finding appropriate treatments, particularly in veterinary practice [48,49]. It could be interesting to more deeply analyze the resistance mechanisms to DHAB in the clones ST1 and ST8. It could be possible that degradation of the dye occurs via the enzyme azoreductase, present in some *S. aureus* strains [66], which is the most extended mechanism. Moreover, some new insights are being investigated concerning the degradation of azo dyes involving new enzymes and pathways, such as a number of oxidases and peroxidases [55,67].

### 2.5. Designing a Therapeutic Utilization of DHAB for Dermatological Application in Veterinary Infections 

The use of topical non-antibiotic treatments could be an interesting alternative to antibiotics in zoonotic infections [68]. Taking account of previous results, the possibility of designing a plaster containing DHAB for cutaneous infections was explored. In the present study, cutaneous plasters have been designed (Figure 7) with a composite material made from sterile agar–water (0.9%) and the azo compound DHAB applied on a sterile commercial plaster. Moreover, other possible composite materials with good biological properties were tested, including gelatin or DHAB entrapped into alginate beads (Appendix A). Unfortunately, both formulations were discarded; the first one because the staphylococci were able to hydrolyze gelatin [65,69] and the composite was not stable. Regarding the second formulation, there was a low diffusion of DHAB from the alginate beads into the growth medium and inhibition of the bacterial growth was not achieved (Appendix A). Thus, the preferred formulation included DHAB in a hydrogel of water–agar (Figure 7A,B).

After 24 h of incubation, growth inhibition halos were observed, particularly for *S. pseudintermedius* (Figure 7C). As expected, the concentration of DHAB needed to obtain inhibition halos in *S. aureus* (Figure 7D) was higher than that needed for *S. pseudintermedius* according to data of MIC (Table 1). Furthermore, the plasters with agar containing DHAB maintained the structure and stability, besides providing a great degree of humidity, which is supposed to be a desirable trait for specific dermal applications such as skin burns or profound wounds.

Our results showed the inhibition of *S. pseudintermedius* growth in vitro upon application of the plaster, besides providing a great degree of humidity necessary for specific dermal applications such as skin burns, profound wounds, or localized bacterial folliculitis such as “hot spots” due to overgrowth of *S. pseudintermedius* at the epidermal level [69]. In this regard, the toxicological profile of DHAB is perfectly known (NCBI, PubChem Compound Summary, 2021). This substance is catalogued as an irritant to the eyes and is harmful if swallowed. However, the suggested application is dermal and the substance is proposed to be solved in an agar base. Moreover, other medical applications of azobenzene derivatives have been recently suggested [21]. The plaster should fulfil several requirements, such as the antibacterial action, nutrition, and moistening of the infected lesion, facilitating the formation of granulation tissue. The possibility of using synergic interactions may be also explored [68], for instance, using a composite material including, besides DHAB, acetylcysteine which is a well-known inhibitor of biofilm formation [70,71]. Other recently described approaches have been the conjugation of five different azo dyes with coumarin to increase antimicrobial activity [72]. In this case, the compounds were able to inhibit the number of CFU.mL^−1^ between 90 and 97%. Approaches such as the one developed in this work may be an easy and affordable alternative for the treatment of skin infections in animals and may contribute to avoiding the crossover use of human antibiotics in livestock, particularly in agricultural communities, which is one of the main causes of the spread of antibiotic resistance [73]. 

## 3. Materials and Methods

### 3.1. Bacterial Strains and Culture Conditions

*Staphylococcus aureus* ATCC29213 and *S. pseudintermedius* LMG 22219 were retrieved from the Spanish Culture Collection (University of Valencia) and from the Belgium Coordinate Collection of Microorganisms, respectively. After obtaining individual colonies on tryptone soy agar (TSA), liquid cultures were prepared in tryptone soy broth (TSB). Frozen cultures were stored at −76 °C in the presence of 15% (*v*:*v*) glycerol. Clinical isolates of *S. aureus* corresponding to clinical samples from 2018 to 2020 were provided by the Reference Laboratory for the Surveillance and Control of Nosocomial Infections and Prudent Use of Antimicrobials Program in Andalucía (PIRASOA program, Hospital Universitario Virgen Macarena, Seville, Spain).

### 3.2. Determination of Bacterial Susceptibility towards Azo Compounds

#### 3.2.1. Antimicrobial Disk Susceptibility Testing

In a first approach, qualitative tests to evaluate the sensitivity of bacterial strains to the compounds were performed on plates of Müeller–Hinton (MH) agar using the test with disks following the protocols described by the CLSI [74]. The plates were streaked with the different bacterial strains using sterile cotton swabs covering the complete surface of the medium in order to obtain confluent growth. A stock solution of the compound DHAB (4,4′-dihydroxyazobenzene) was prepared in water at 20 mg mL^−1^ (as a suspension, since the compound was insoluble at this concentration). Sterile blank paper disks were loaded with 15 µL of the suspension and allowed to dry at room temperature, with a control disk with water being kept. The disks were placed onto the surface of the medium and the plates were sealed and incubated at 37 °C for 48 h. After this time, the plates were observed for the presence of inhibition halos around the disks and the diameter of the inhibition halos was recorded.

#### 3.2.2. Determination of MIC and MBC

MIC of DHAB was determined by microdilution assay according to [75]. Microdilution was performed in 96-well plates containing concentrations of DHAB (prepared in glycerol) between 512 and 0.5 mg L^−1^ in MHB medium (Invitrogen, Spain). For the determination of the MIC, a stock solution of DHAB was prepared at a concentration of 5.12 g L^−1^ in glycerol as a biocompatible solvent. For that, the solutions were sonicated at 50/60 Hz and 100% amplitude for 2 min using a Hielscher Ultrasound Technology UP400S. The determination of the MIC was completed by a microdilution test in 96-well microtiter plates. Aliquots of 200 µL of Müeller–Hinton Broth medium (MHB) were supplemented with different concentrations of DHAB from 512 mg L^−1^ to 0.5 mg L^−1^ (a serial dilution following a base-2 logarithmic gradient of concentrations) and deposited in the wells of the microtiter plate in triplicate. Wells containing no DHAB were kept as controls (with only MHB medium). The wells were inoculated with 5 µL of overnight cultures of *S. aureus* or *S. pseudintermedius* (the optical density at 600 nm of the cultures had been previously adjusted to 1.0 with sterile MHB). A non-inoculated row was also used as a control of sterility. The plate was sealed with film tape. After incubating the plate for 24 h at 37 °C, turbidity was visually observed and also determined by measuring the optical density at 600 nm in a microtiter plate reader ASYS UVM340. The MIC of DHAB (mean of three determinations) was considered as the lowest concentration that showed no visible growth.

For the determination of the MBC, 100 µL of the clear wells (without apparent growth) were spread on TSA. After incubating the plates for 48 h at 37 °C, the MBC corresponded to the plate where no growth of colonies was observed.

### 3.3. Determination of ROS-Scavenging Enzymes

*S. aureus* ATCC 29213 and *S. pseudintermedius* LMG 22219 were grown in 50 mL TSB for 24 h at 37 °C and divided into five aliquots of 10 mL each. One of them was kept as a control and the other four were supplemented with DHAB at MIC (in duplicate) and MBC (in duplicate). Aliquots were grown for additional 24 h for indication of ROS-scavenging enzymes. 

After centrifuging the cultures at 8000 rpm for 5 min, pellets were resuspended in 2.5 mL of extraction buffer containing 50 mM potassium phosphate, pH 7, and 2 mM EDTA. Tubes containing the cell suspensions were placed in an ice bath. Cell disruption was performed by sonication using an Ultrasonic Processor (Hielscher) with amplitude 100% and cycle 0.8. The cells were submitted to three periods of 30 s alternating with three periods of 1 min for cooling. After centrifuging the cell homogenates at 10,000 rpm for 10 min at 4 °C, the supernatants were used as crude extracts for enzymes and protein determination.

Catalase activity (CAT) was determined at room temperature by following the decrease in the absorbance at 240 nm due to the disappearance of H_2_O_2_ using a Perkin Elmer Lambda 25 UV/Vis spectrophotometer (Shelton, CT, USA) and a value of ε = 39.4 mM^−1^ cm^−1^ [76]. Total peroxidases were determined according to [77] by measuring the absorbance at 420 nm due to the oxidation of pyrogallol to purpurogallin by H_2_O_2_ (ε = 12 mM^−1^ cm^−1^). Additionally, superoxide dismutase activity (SOD) was determined by measuring the absorbance at 560 nm due to the inhibition of the photoreduction of nitroblue tetrazolium (NBT) in the presence of riboflavin. One SOD unit was the amount of enzyme able to inhibit 50% of NBT photoreduction by riboflavin [78].

The enzymatic activities were relativized to the protein content of the crude extracts, which were determined according to a calibration curve made with bovine serum albumin fraction V (Sigma) following the previously described method [79]. 

### 3.4. Determination of Damage to Membranes Based on Thiobarbituric Acid Reactive Substances (TBARS)

For the determination of thiobarbituric acid reactive substances (TBARS) such as malondialdehyde (MDA), cultures of 50 mL of *S. aureus* or *S. pseudintermedius* were cultivated for 48 h in the presence of TSB at 37 °C and 200 rpm as described before. The culture of each strain was divided into five aliquots of 10 mL each and cultivated for an additional 24 h, either in TSB (control) or in the presence of DHAB at MIC (in duplicate) or MBC (in duplicate). After growth cultures were pelleted by centrifugation at 8000 rpm for 5 min, the supernatants were discarded and the bacterial pellets were resuspended in 3 mL of 20% trichloroacetic acid (TCA) containing 0.5% thiobarbituric acid (TBA) [80]. The cell suspensions were extracted for 30 min at 95 °C in a water bath followed by rapid chilling in an ice bath. After extraction, the homogenates were centrifuged at 8000 g for 5 min and the absorbance at 532 nm was measured using a PerkinElmer Lambda 25 UV/Vis spectrophotometer (PerkinElmer, Shelton, Connecticut, USA) and a molar extinction coefficient of ε = 155 mM^−1^ cm^−1^.

### 3.5. Effect of DHAB on Biofilm Formation

A preliminary evaluation of the effect of DHAB on biofilm formation was completed by a colorimetric assay using microtiter plates according to [81]. Cultures of *S. aureus* and *S. pseudintermedius* were grown overnight in TSB medium at 37 °C and 200 rpm. The columns of microtiter plates were filled with 200 µL of TSB containing decreasing concentrations of DHAB from 512 to 0.5 mg L^−1^ following a base-2 logarithmic gradient. A triplicate repetition of each concentration was performed. A final row with TSB without DHAB was kept as a control of biofilm formation in the absence of DHAB. The wells of the plate were inoculated with 5 µL of the cultures of *S. aureus* or *S. pseudintermedius* (absorbance at 600 nm adjusted to 1.0). The plate was sealed with tape film and incubated for 48 h at 37 °C. After incubation, the plate was inverted and shaken vigorously in order to empty the wells, which were washed thrice with 250 µL of sterile distilled water. The plate was allowed to dry at room temperature for 30 min and the biofilm formed at the bottom of the wells was stained with 200 µL of 1% crystal violet for 20 min. After voiding the plate by inversion and shaking, the wells were washed thrice with sterile distilled water and allowed to dry as before. Finally, the dye was suspended in 200 µL of a solution of acetic acid: ethanol (1:2 *v*:*v*) and incubated for 30 min at room temperature. The absorbance at 570 nm in comparison with non-inoculated wells was registered using a microtiter plate reader ASYS UVM340. 

In order to confirm the inhibition of biofilm formation, direct observation by scanning electron microscopy (SEM) was performed. Biofilms were allowed to form onto the surface of 1 cm-diameter glass thin slides. For this, individual slides were deposited in the wells of 24-wells polystyrene plates (in duplicate). The wells were filled with 5 mL of TSB supplemented either with or without DHAB at MIC or MBC (enough volume to fully cover the glass slide). The wells were inoculated with 0.1 mL of overnight cultures of *S. aureus* or *S. pseudintermedius* and allowed to grow for 4 days at 28 °C. After the formation of biofilms, the slides were transferred face-up to a clean 24-wells polystyrene plate and washed thrice with sterile distilled water and allowed to dry at room temperature. For the fixation of bacteria, the slides were treated with 5 mL of 2.5% glutaraldehyde prepared in 0.2 M cacodylate buffer pH 7.2 for 3 h at room temperature, followed by three washes in 5 mL of 0.2 M cacodylate buffer pH 7.2. Dehydration was performed in acetone series (from 50% to pure acetone). The samples were dried using a critical point drier Leica EM CPD300 at 31 °C and 73.8 bar, sputtered with Au-Pd (10 nm) and observed with a scanning electron microscope Zeiss EVO LS15 at the General Research Services of the University of Sevilla (CITIUS). 

### 3.6. PFGE Electrophoresis

The genetic relationship between clinical isolates was assessed by pulsed field gel electrophoresis (PFGE). Chromosomal DNA was digested using the enzyme *SmaI* and the fragments were separated using a CHIEF DR-II system (Biorad, Madrid, Spain). The gels were run under the same voltage and time conditions (6 volts at an angle of 120°) and the pulses for *S. aureus* were 5–15 pulses for 10 hours and then 15–45 pulses for 13 hours. The normalization strain used was *S. aureus* NCTC8325 in order to be able to compare the gels that have not been run in parallel. Banding patterns of the gel were analyzed using Bionumerics 7.6 software (AppliedMaths, Austin, TX, USA). An unweighted pairwise clustering method with an arithmetic mean clustering algorithm (UPGMA) was used to generate a dendrogram and Dice’s coefficient to measure the genetic similarity between the isolates were set to 1%. Isolates that differed in two or more bands in the PFGE analysis were assigned to different pulse types [82]. 

### 3.7. Statistical Analysis

The determination of the MIC and MBC was performed in triplicate. Enzymatic activities and MDA were the averages ± standard deviations of three determinations in two independent cultures. The formation of biofilms onto the base of wells of microtiter plates was performed in triplicate and data are means ± standard deviations of three determinations. Comparison of the means was completed using the Student’s t test and significant differences at *p* < 0.05 are indicated in the figures and tables. The observation of biofilms by SEM was performed in duplicate. 

## 4. Conclusions

In this study, the antimicrobial properties of 4,4′-dihydroxyazobenzene (DHAB) towards *S. aureus* and *S. pseudintermedius* are described. The values of MIC and CMB were similar to other azo compounds described in the literature. However, for Gram-negative microorganisms, specifically *Escherichia coli* and *Pseudomonas aeruginosa*, it has not exhibited convenient antimicrobial activity. Moreover, the substance has been tested for ten clones of *S. aureus* isolated from valuable samples (eight methicillin-resistant clones and two daptomycin-resistant clones); 80% of them showed the same susceptibility towards DHAB as the control strain.

The mechanisms of action of this substance have been investigated in detail at the cellular level. On the one hand, it causes oxidative stress based on the increase in different antioxidant enzymes (peroxidases or superoxide dismutase depending on the bacterial species). DHAB also causes damage to the membrane by peroxidation of its lipids. Together with the antimicrobial activity and oxidative stress, it has been possible to verify the action of this substance against the formation of bacterial biofilms, which is a key virulence factor in pathogens that favors resistance to antibiotics and chronification of infections.

Finally, a topical therapeutic application has been successfully designed using a composite material, allowing for a completely biocompatible, economical, and effective option against the *Staphylococcus* genus.

## Figures and Tables

**Figure 1 antibiotics-11-01800-f001:**
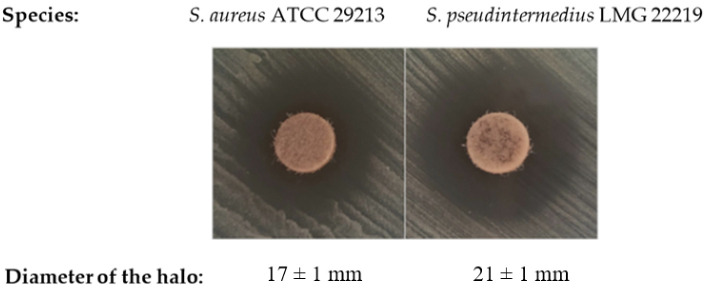
Semi-quantitative analysis of the inhibitory effect of DHAB on the growth of *Staphylococcus* spp. Using the disks test. Data are mean ± standard deviation of three determinations.

**Figure 2 antibiotics-11-01800-f002:**
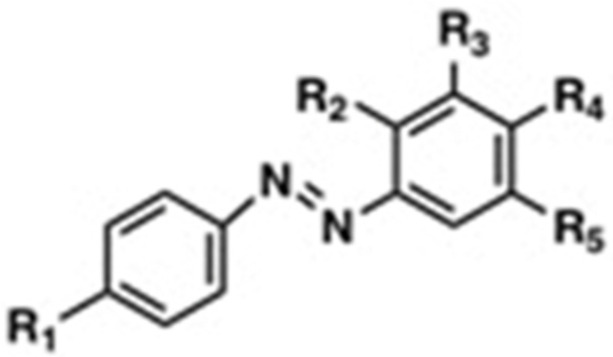
Structure of azo compounds tested as antimicrobial substances.

**Figure 3 antibiotics-11-01800-f003:**
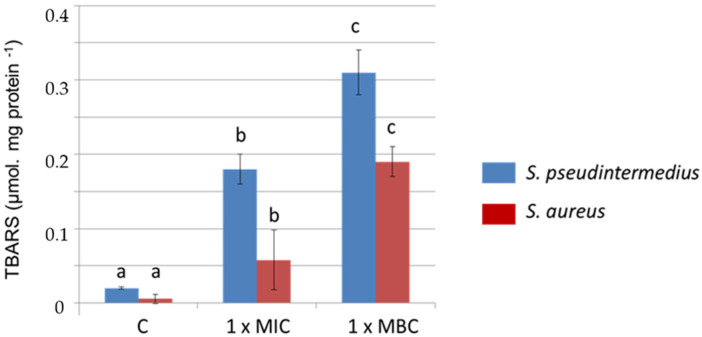
Levels of thiobarboturic acid reactive substances (TBARS) in cells of *S. pseudintermedius* and *S. aureus* treated with DHAB for 24 h at two different doses: MIC and MCB. C: control cells grown in TSB. Data are means of three determinations in two independent cultures, and significant differences at *p* < 0.05 are indicated by different letters; (a) correspond to the level of MDA in the strains cultivated in the absence of DHAB; (b, c) indicate significant differences in the presence of DHAB.

**Figure 4 antibiotics-11-01800-f004:**
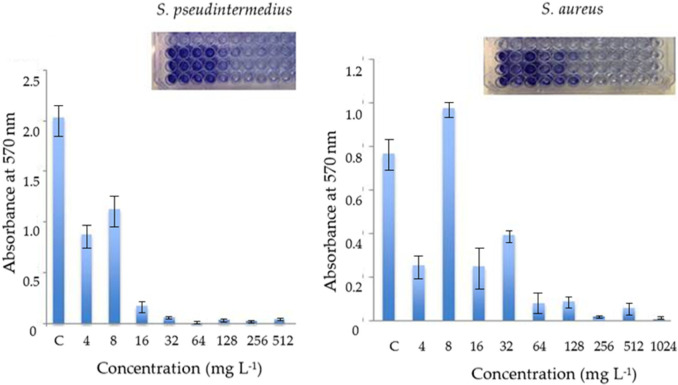
Effect of DHAB on the formation of biofilms in vitro by *S. pseudintermedius* and *S. aureus*. Data are means ± standard deviations of three independent determinations.

**Figure 5 antibiotics-11-01800-f005:**
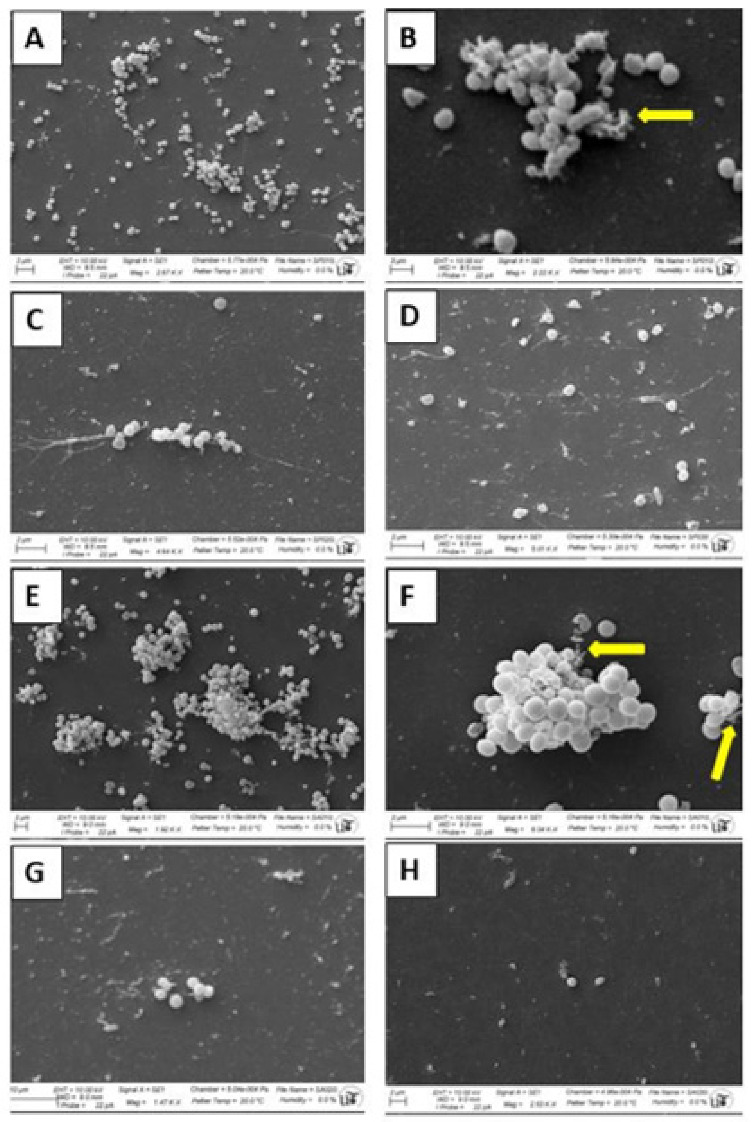
Observations by SEM of biofilms formed by staphylococcal species after 24 h on glass slides and the effect of DHAB on biofilm formation. (**A**) Biofilm formed by *S. pseudintermedius* in the absence of DHAB; (**B**) closer view of colonies of *S. pseudintermedius* in the absence of DHAB. Yellow arrows point out extracellular material (particulated material and fibers) secreted by bacteria for attachment; (**C**) small colonies of *S. pseudintermedius* in the presence of DHAB at MIC; (**D**) individual cells of *S. pseudintermedius* in the presence of DHAB at MBC. Note the presence of small and deformed cells; (**E**) biofilm formed by *S. aureus* in the absence of DHAB; (**F**) colonies of *S. aureus* in the absence of DHAB. Yellow arrows show extracellular material (particulated material and fibers) secreted by bacteria for attachment; (**G**) small colonies of *S. aureus* in the presence of DHAB at MIC; and (**H**) individual cells of *S. pseudintermedius* in the presence of DHAB at MBC. Note the presence of small and deformed cells at a high DHAB concentration.

**Figure 6 antibiotics-11-01800-f006:**
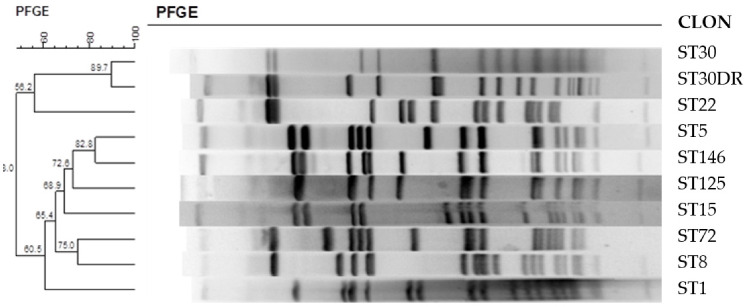
Analysis of the genetic diversity of ten clinically relevant isolates of *Staphylococcus aureus* by PFGE.

**Figure 7 antibiotics-11-01800-f007:**
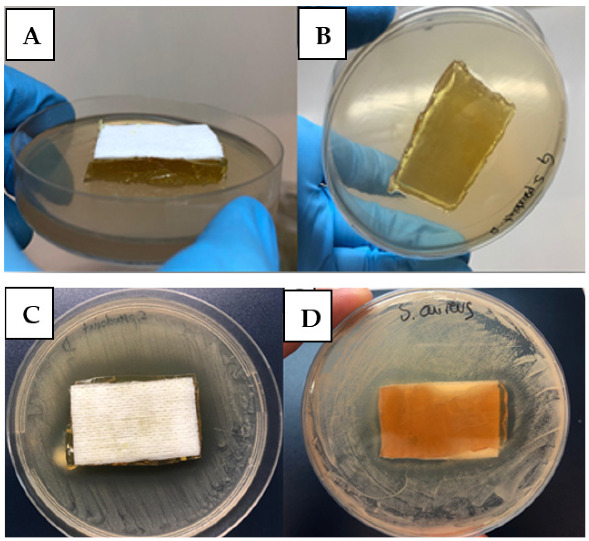
(**A**) Dermal sterile cotton plaster loaded with composite material made from 0.9% agar containing DHAB. (**B**) Aspect of the plaster from the bottom of the plate. (**C**) Growth inhibition halo observed for *S. pseudintermedius* after 24 h of incubation at 37 °C. (**D**) Growth inhibition observed for *S. aureus* after 24 h of incubation at 37 °C.

**Table 1 antibiotics-11-01800-t001:** Values of MIC and MBC for *Staphylococci* towards DHAB.

Bacteria	Parameter	
*Staphylococcus aureus* ATCC 29213	MIC	64 mg L^−1^
	MBC	256 mg L^−1^
*Staphylococcus pseudintermedius* LMG 22219	MIC	32 mg L^−1^
	MBC	64 mg L^−1^

**Table 2 antibiotics-11-01800-t002:** MIC for different azo compounds against *E. coli* and *S. aureus*. N.D., not determined.

Name of the Compound	R1	R2	R3	R4	R5	MIC (mg mL^−1^)		Reference
						*E. coli*	*S. aureus*	
4-((4-Hydroxyphenyl)4iazinyl) benzene-1,3-diol	OH	OH	H	OH	H	>128	>128	Piotto et al., 2017 [25]
4-(p-Tolyldiazenyl) benzene-1,3-diol	CH_3_	OH	H	OH	H	>32	>32	Piotto et al., 2017 [25]
4-((4-Methoxyphenyl)4iazinyl) benzene-1,3-diol	OCH_3_	OH	H	OH	H	>128	16	Piotto et al., 2017 [25]
4′-Hydroxyazobenzene	OH	H	H	H	H	N.D.	25	Concilio et al., 2017 [35]
4′-Hydroxy-4- methoxyazobenzene	OH	H	H	CH_3_	H	N.D.	25	Concilio et al., 2017 [35]
4′-Hydroxy-4- Methylazobenzene	OH	H	H	OCH_3_	H	N.D.	20	Concilio et al., 2017 [35]
**DHAB**	**OH**	**H**	**H**	**OH**	**H**	**>512**	**64**	**This work**

**Table 3 antibiotics-11-01800-t003:** EUCAST Breakpoint Table for MIC of *Staphylococcus aureus* ATCC 29213 β-lactamase-producing strain (weak).

Antimicrobial Agent	MIC (mg L^−1^)
Ciprofloxacin	0.25
Gentamicin	0.25–0.5
Erythromycin	0.5
Vancomicin	1
Fosfomycin	1–2
Amikacin	2
Cefoxitin	2
Chloramphenicol	4–8
Nitrofurantoin	16
**DHAB**	**64**

**Table 4 antibiotics-11-01800-t004:** Determination of ROS-scavenging enzymes in *S. pseudintermedius* and *S. aureus* in the presence of DHAB at concentrations 1 × MIC and 1 × MBC. Data are means ± standard deviations of three independent determinations in two independent cultures. Significant differences at *p* < 0.05 are indicated by different letters (^a, b, c^). Data in bold correspond to significant inductions.

Strain	Enzyme	Activity (mU mg^−1^ Protein)		
		Control	DHAB at MIC	DHAB at MBC
*S. pseudintermedius* LMG 22219	Catalase	0.75 ± 0.12 ^(a)^	0.38 ± 0.06 ^(b)^	0.66 ± 0.04 ^(a)^
	Total Peroxidase	26 ± 2 ^(a)^	4.0 ± 1.0 ^(b)^	21.0 ± 6.0 ^(a)^
	Superoxide dismutase	11.09 ± 1.33 ^(a)^	12.60 ± 1.50 ^(a)^	**23.75 ± 1.93 ^(b)^**
*S. aureus* ATCC 29213	Catalase	0.74 ± 0.14 ^(a)^	0.44 ± 0.07 ^(b)^	0.84 ± 0.06 ^(a)^
	Total Peroxidase	25.0 ± 6.0 ^(a)^	2.6 ± 0.4 ^(b)^	**70.0 ± 14.0 ^(c)^**
	Superoxide dismutase	4.90 ± 0.50 ^(a)^	3.94 ± 0.30 ^(a)^	5.09 ± 0.52 ^(a)^

(^a, b, c^) Significant differences at *p* < 0.05 are indicated by different letters with regard to the activity in the absence of DHAB.

**Table 5 antibiotics-11-01800-t005:** Provenance of ten clinically relevant *Staphylococcus aureus* isolates from different clinical samples at the hospital Virgen Macarena (Sevilla, Spain).

Sample Identification	Clinical Sample	CLON	Determinant	CMI (mg L^−1^)	CMB (mg L^−1^)
SA1	Blood	ST30	MRSA	64	256
SA2	Swab	ST30 DR	MSSA (DAPTO R)	64	256
SA3	Respiratory tract	ST22	MRSA	64	256
SA4	Swab	ST5	MSSA (DAPTO R)	64	256
SA5	Injury swab	ST146	MRSA	64	256
SA6	Swab	ST125	MRSA	64	256
SA7	Nasal smear	ST15	MRSA	64	256
SA8	Nasal smear	ST72	MRSA	64	256
SA9	Swab	ST8	MRSA	256	>512
SA10	Respiratory tract	ST1	MRSA	256	>512
ATCC 29213			MSSA	64	256

## Data Availability

Not applicable.

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
