# Peer review of "Antimicrobial and Antibiofilm Effect of 4,4′-Dihydroxy-azobenzene against Clinically Resistant Staphylococci"

_antibiotics, 2022, doi:10.3390/antibiotics11121800_

Round 1
Reviewer 1 Report
In the presented manuscript, Pérez-Arandaand co-workers analyzed “Antimicrobial and antibiofilm effect of 4,4’-dihydroxyazobencene against clinical resistant staphylococci”.
The manuscript is suitable for publication in Antibiotics, the language should be revised and additional analyses should be performed to present a comprehensive analysis as proposed by the authors.
I provide some major and minor comments below.
Major;
1. Line 145. Why didn't the authors test other gram-negative species?
2. Repeats information: 141-145 and 176-179
3. The authors do not mention the number of replicates they used for the analysis?
4. Why did the authors use PFGE and not rep-PCR or ERIC-PCR?
5. PFGE analysis of S. pseudintermediusis not reported.
6. Fig.6. The gel appears to be cut in each of the rails. Wasn't it run parallel?
Minor,
The writing and grammar of the article must be revised.
1. Line 66: when fighting?
2. Fig. 5: Please, increase the resolution; the information under the image is not visible.
Author Response
RESPONSES TO REVIEWERS
Dear Reviewers,
We would like to thank you for the comments to improve our paper entilted «Antimicrobial and antibiofilm effect of 4,4’-dihydroxyazobenzene against clinical resistant staphylococci » by Dr. Pérez-Aranda et al.
After carefully reading all the valuable suggestions, we have elaborated a detailed one-by-one response to all the items. We have highlighted in yellow the corresponding changes in the revised version of the manuscript. Moreover, the text highlighted in blue corresponds to changes made in the writing of some paragraghs of Materials and Methods for high coincidence upon suggestion of the Editor (we have fully re-written the Materials and Methods section). In our opinion, the changes done have substantially improved the quality and relevance of the paper and, thus, we hope that these changes and corrections are adequate for publishing our work in Antibiotics.
REVIEWER 1
In the presented manuscript, Pérez-Aranda and co-workers analyzed “Antimicrobial and antibiofilm effect of 4,4’-dihydroxyazobenzene against clinical resistant staphylococci”.
The manuscript is suitable for publication in Antibiotics, the language should be revised and additional analyses should be performed to present a comprehensive analysis as proposed by the authors.
Thank you very much for your positive comment. We have carefully revised English and we have introduced the corresponding changes to major and minor comments.
I provide some major and minor comments below.
Major;
- Line 145. Why didn't the authors test other gram-negative species?
Authors do appreciate this comment. The four species initially selected to evaluate the antimicrobial effect of DHAB were S. aureus, E. coli, P. aeruginosa and, due to the relevance to cutaneous infections in pets in the veterinary practice, S. pseudintermedius. The three former species are the ones commonly used in most of the papers for the testing of new compounds with putative antimicrobial activity (Haque et al., 2014; Montero et al., 2019). The initial evaluation of the susceptibility to DHAB showed that Gram positive were sensible to this compound but on the contrary, Gram negative showed not desirable effect. Other previously studied azo compounds (such as the ones described in Piotto et al., 2017 and Concilio et al., 2017) were also useful against Gram positive but not against Gram negative. Our results, together with these previous findings, along with the fact that Gram negative can degrade many azo dyes quite effectively, led us to stop investigation against Gram negative. In Supplementary Information Figure S1, we provided data showing that both Gram negative species can effectively use the azo compound DHAB as the sole C source, so it means that they are able, not only to tolerate it, but to degrade it and use it as a source of carbon. In this regard, several Gram negative species such as Pseudomonas fluorescens or Enterobacter hormaechei have been proposed for bioremediation of azo dyes pollution by textile industries, since they are able to fully degrade these compounds (Zabłocka-Godlewska et al., 2014; Thangaraj et al., 2021).
Haque S, Nawrot DA, Alakurtti S, Ghemtio L, Yli-Kauhaluoma J, et al. (2014) Screening and Characterisation of Antimicrobial Properties of Semisynthetic Betulin Derivatives. PLoS ONE 9(7): e102696. doi:10.1371/journal.pone.0102696.
Montero, D.A., Arellano, C., Pardo, M. et al. Antimicrobial properties of a novel copper-based composite coating with potential for use in healthcare facilities. Antimicrob Resist Infect Control 8, 3 (2019). https://doi.org/10.1186/s13756-018-0456-4.
Zabłocka-Godlewska E, Przystaś W, Grabińska-Sota E. Decolourisation of Different Dyes by two Pseudomonas Strains Under Various Growth Conditions. Water Air Soil Pollut. 2014;225(2):1846. doi: 10.1007/s11270-013-1846-0. Epub 2014 Jan 22. PMID: 24578585; PMCID: PMC3928507.
Sheela Thangaraj, Paul Olusegun Bankole, Senthil Kumar Sadasivam. 2021. Microbial degradation of azo dyes by textile effluent adapted, Enterobacter hormaechei under microaerophilic condition. Microbiological Research, 250: 126805. https://doi.org/10.1016/j.micres.2021.126805.
- Repeats information: 141-145 and 176-179
Authors are grateful and would like to point out that, when the information in line 176 is commented again, is because we are doing a discussion and comparison of the results in different strains. We also discuss the mechanisms of degradation of DHAB by gram negative. Nevertheless, the text is not literally the same, although the idea is discussed, first as a Result, and secondly as Discussion.
- The authors do not mention the number of replicates they used for the analysis?
Thank you for the comment. This information has been actualized in the section “3.7. Statistical analysis”. Three replicates of each determination have been performed (for MIC, MBC, biofilms formation, ROS-scavenging enzymes and MDA content). Two replicates were done for the observation of biofilms by SEM.
- Why did the authors use PFGE and not rep-PCR or ERIC-PCR?
We agree with the reviewer that methods other than PFGE can be used for molecular typing and assessment of diversity among the different isolates. However, PFGE is the "gold standard" and one of the most used methods for typing bacteria (Neoh et al., 2019). That is why we used PFGE and not other techniques. Nevertheless, our aim was only to demonstrate the diversity of the S. aureus isolates, which were obtained from different clinical samples and different patients in a period of 2 years (2018-2020). We just wanted to highlight the diversity of the collection, and in fact, this could have been done by other techniques such as BOX-PCR, rep-PCR, or ERIC-PCR, RFLP, etc.
Neoh HM, Tan XE, Sapri HF, Tan TL. Pulsed-field gel electrophoresis (PFGE): A review of the "gold standard" for bacterial typing and current alternatives. Infect Gene Evol. 2019 Oct;74:103935. doi: 10.1016/j.meegid.2019.103935. Epub 2019 Jun 22. PMID: 31233781.
- PFGE analysis of S. pseudintermediusis not reported.
Thank you very much for the comment. Unfortunately, we de not have a collection of isolates from S. speudintermedius which is mainly an animal pathogen causing skin infections. However, the importance of this pathogen is that, more and more frequently, it causes infections in humans due to close interaction with pets or other animals. In the hospital Virgen Macarena, in the Unit of Infectious Diseases and Clinical, there was a wide collection of S. aureus strains, because it is one of the most prevalent human pathogens found in hospitals. But such a collection of S. pseudintermedius strains is not available for us. It could be very interesting to do the analysis suggested by the reviewer in collaboration with veterinarian clinics, which most problaby can provide a wide and diverse collection of this microorganism.
- Fig.6. The gel appears to be cut in each of the rails. Wasn't it run parallel?
Thank you very much for the comment. It is a common practice at the hospital to evaluate the diversity of the isolates together with the susceptibility to several antibiotics. For this purpose, we routinely use PFGE. However, many times, not all the strains are run in the same gel. In order to be able to compare the band patterns of independent runs, a normalization strain is always introduced in the gels. In this regard, it is not necessary for molecular typing to run the gels in parallel. For this, the normalization strain used in the case of S. aureus is the strain NCTC8325, to be able to compare gels that have not been run in parallel.
Minor,
The writing and grammar of the article must be revised.
English has been revised.
- Line 66: when fighting?
We have changed the expression by « at the time of treating these infections »
- Fig. 5: Please, increase the resolution; the information under the image is not visible.
Unfortunately, Fig. 5 was made with the original pictures provided by the SEM microscope. We cannot change these pictures, but we have edited them and tried to ameliorate the resolution.
Reviewer 2 Report
Dear Authors,
I really like presented research plan, but not the object od investigation.
DHAB should be considered as well antimicrobial. In my opinion, this compound is not "lead" structure that could be improved.
Major comments: In some cases, three is a lack of reference antibiotic against used bacterial strain. All results and figures must be corrected.
No novel structures presented. Most of investigation is linked to one compound.
Minor comment: DHAB name in whole text must be corrected. B stands for benzene not bencene.
Please do adjustments in manuscript. All to all, I believe this paper might be interesting to Antibiotics readers.
Author Response
REVIEWER 2
Dear Authors,
I really like presented research plan, but not the object of investigation.
Thank you very much for your comments and the appreciation of the research plan. In our hands, the investigation object could be relevant due to the terapeutic limitations for the veterinary practice. The discovery of non-toxic substances with putative antimicrobial activity that can be used in animals sking infections can be an alternative in the veterinary practice when the treatment with human-reserved antibiotics is forbiden. Moreover, as we will discuss later, it is important to investigate the mechanism of action of prospected new antimicrobial compounds.
DHAB should be considered as well antimicrobial. In my opinion, this compound is not "lead" structure that could be improved.
Thank you very much for the comment. In fact, several compunds based on the structure of DHAB (in which the –H have been substituted by different radicals) have been described (Piotto et al., 2017 ; Concilio et al., 2017). In this particular, the azo structure can be a lead structure for the design of novel azo compounds by derivatization. As depicted in Table 2, some of them previously reported by other authors displayed activity against Gram positive but not against E. coli.
Piotto S.; Concilio S.; Sessa L.; Diana R.; Torrens G.; Juan C.; et al. Synthesis and Antimicrobial Studies of New Antibacterial Azo-Compounds Active against Staphylococcus aureus and Listeria monocytogenes. Molecules 2017, 22(8), 1372. Doi: 10.3390/molecules22081372.
Concilio, S.; Sessa, L.; Petrone, A.M.; Porta, A.; Diana, R.; Iannelli, P.; et al. Structure Modification of an Active Azo-Compound as a Route to New Antimicrobial Compounds. Molecules 2017, 22(6), pii: E875. Doi: 10.3390/molecules22060875.
Major comments: In some cases, three is a lack of reference antibiotic against used bacterial strain. All results and figures must be corrected.
Thank you for the observation. We have introduced new information to avoid this problem. For comparison with our data, in adition to Table 2 where we have compared DHAB with other azo compounds, we have now introduced Table 3, where we compare our data with the MIC for several antibiotics for the type strain S. aureus ATCC29213 ; these data are available through the European Commitee for Antimicrobial Susceptibility Testing (EUCAST). It is a document to establish quality control when performing MIC tests.
No novel structures presented. Most of investigation is linked to one compound.
Thank you for the comment. In fact, the aim of our study was to analyze the antimicrobial properties of just this representative azo compound. Since other azo compounds of similar structure had been previously studied (Piotto et al., 2017; Concilio et al., 2017), our study has been more focused on the description of the susceptibility for two staphylococci and the study of the mechanisms of action, in particular the antimicrobial and antibiofilm activities. We have also described the induction of oxidative stress by this substance and the damage to membranes caused by it. Such deep analyses had not been reported before with other azo compounds. We have preferred to do a more profound study of the mechanism instead of using a battery of azo compounds with similar structure and different substituents. Nevertheless, as the reviewer says, the possibility of extending this study to new azo compounds can be assessed in future work. In addition, the proposal of low-cost cutaneous plasters for skin infections have not been previously reported and can have a direct application in the veterinary practice.
Minor comment: DHAB name in whole text must be corrected. B stands for benzene not bencene.
Thank you so much, the mistake has been corrected.
Please do adjustments in manuscript. All to all, I believe this paper might be interesting to Antibiotics readers.
Thank you very much for your comment. We have tried to improve both the presented results and English in order to make our research more attractive to Antibiotics readers.
Round 2
Reviewer 1 Report
The authors could include the information in the bottom of figures a,b,c,d,e,f,g,h independently, because there is no way for the reader to distinguish the information shown.
The fact that the authors mention that this is a common practice in the hospital does not indicate that it is correct. Genetic diversity analyses have to be run in parallel, the reader will not know if each of the gel tracks shown was run at a different voltage or time.
Author Response
RESPONSES TO REVIEWERS
Dear Reviewer 1,
We would like to thank you for the comments to improve our paper entilted «Antimicrobial and antibiofilm effect of 4,4’-dihydroxyazobenzene against clinical resistant staphylococci » by Dr. Pérez-Aranda et al.
After carefully reading all the valuable suggestions, we have elaborated a detailed one-by-one response to the items. We have highlighted in yellow the corresponding changes in the revised version of the manuscript. We hope that these changes and corrections are adequate for publishing our work in Antibiotics.
REVIEWER 1
The authors could include the information in the bottom of figures a,b,c,d,e,f,g,h independently, because there is no way for the reader to distinguish the information shown.
The legend of Fig. 5 has been fully re-written and every part of the figure has been explained individually.
The fact that the authors mention that this is a common practice in the hospital does not indicate that it is correct. Genetic diversity analyses have to be run in parallel, the reader will not know if each of the gel tracks shown was run at a different voltage or time.
We agree with the reviewer that most typing methods have to be run in parallel in order to compare the band patterns of different isolates. However, in the clinical practice, you can rarely run the different isolates on the same gel or on gels in parallel. In fact, the PFGE is made for typing bacteria from sprouts and for comparison over the years, if necessary. The gels are always run under the same voltage and time conditions (6 volts at an angle of 120 º) and the pulses for S. aureus are 5-15 pulses 10 hours and then 15-45 pulses 13 hours. An image is then generated, which is further processed by a program that normalizes the PFGE. Thus, the final image is generated with a program, BIONUMERICS, which processes the image and normalizes it (https://www.applied-maths.com/bionumerics/modules/fingerprint-data-module). We provide two previously published references, one of them by Machuca et al., 2022 in which PFGE is used for the molecular typing of an outbreak over time ; and another one by Mentula et al., 2006, in which 400 E. coli isolates are analyzed by PFGE (which is impossible for them to fit in a gel).
Machuca, J., Lopez-Cerero, L., Rodríguez-Maresca, M., Fernández-Cuenca, F., López-Hernández, I., Delgado-Valverde, M., ... & Pascual, A. (2022). Molecular characterisation of an outbreak of NDM-7-producing Klebsiella pneumoniae reveals ST11 clone expansion combined with interclonal plasmid dissemination. International Journal of Antimicrobial Agents, 59(4), 106551.
Mentula, S., Virtanen, T., Kanervo-Nordström, A., Harmoinen, J., Westermarck, E., Rautio, M., ... & Könönen, E. (2006). Relatedness of Escherichia coli strains with different susceptibility patterns isolated from beagle dogs during ampicillin treatment. International journal of antimicrobial agents, 27(1), 46-50.
Reviewer 2 Report
Authors reply is satisfying.
Author Response
Thank you very much.
Round 3
Reviewer 1 Report
The authors made corrections.
Author Response
Dear Reviewer,
Please find attached the revised version of the manuscript entitled «Antimicrobial and antibiofilm effect of 4,4’-dihydroxyazobenzene against clinical resistant staphylococci » by Dr. Pérez-Aranda et al.
We have performed the following changes in the manuscript:
- PFGE electrophoresis running conditions and the normalization control strain should be included in the M&M section.
- Some English spelling errors have been corrected.
We have highlighted in yellow the corresponding changes in the revised version of the manuscript. We hope that these changes and corrections are adequate for publishing our work in Antibiotics.
Best regards,
Eloísa Pajuelo and Ana Alcudia (corresponding authors)